# STING Agonists as Cancer Therapeutics

**DOI:** 10.3390/cancers13112695

**Published:** 2021-05-30

**Authors:** Afsaneh Amouzegar, Manoj Chelvanambi, Jessica N. Filderman, Walter J. Storkus, Jason J. Luke

**Affiliations:** 1Department of Medicine, University of Pittsburgh, Pittsburgh, PA 15213, USA; amouzegara@upmc.edu; 2Department of Immunology, University of Pittsburgh, Pittsburgh, PA 15213, USA; mac392@pitt.edu (M.C.); JEF116@pitt.edu (J.N.F.); storkuswj@upmc.edu (W.J.S.); 3UPMC Hillman Cancer Center, Pittsburgh, PA 15232, USA

**Keywords:** stimulator of interferon genes, cGAS, STING agonist, type I interferon, tumor vasculature, anti-tumor immunity, drug delivery

## Abstract

**Simple Summary:**

Immunotherapies have revolutionized the field of cancer therapeutics, yet a substantial subset of patients fail to respond. Recent efforts have focused on identifying targets that could elicit or augment anti-tumor immune responses. One such novel target is STING or stimulator of interferon (IFN) genes, an endoplasmic protein that induces the production of pro-inflammatory cytokines such as type I IFNs. Since the discovery of STING, numerous natural and synthetic STING agonists have been tested in both pre-clinical and clinical settings in different tumors. However, the structural instability of first-generation agonists prompted the development of more stable and potent compounds. This review will highlight the latest pharmacologic classes of STING agonists, novel approaches for tumor-targeted drug delivery, and challenges in the clinical targeting of the STING pathway.

**Abstract:**

The interrogation of intrinsic and adaptive resistance to cancer immunotherapy has identified lack of antigen presentation and type I interferon signaling as biomarkers of non-T-cell-inflamed tumors and clinical progression. A myriad of pre-clinical studies have implicated the cGAS/stimulator of interferon genes (STING) pathway, a cytosolic DNA-sensing pathway that drives activation of type I interferons and other inflammatory cytokines, in the host immune response against tumors. The STING pathway is also increasingly understood to have other anti-tumor functions such as modulation of the vasculature and augmentation of adaptive immunity via the support of tertiary lymphoid structure development. Many natural and synthetic STING agonists have entered clinical development with the first generation of intra-tumor delivered cyclic dinucleotides demonstrating safety but only modest systemic activity. The development of more potent and selective STING agonists as well as novel delivery systems that would allow for sustained inflammation in the tumor microenvironment could potentially augment response rates to current immunotherapy approaches and overcome acquired resistance. In this review, we will focus on the latest developments in STING-targeted therapies and provide an update on the clinical development and application of STING agonists administered alone, or in combination with immune checkpoint blockade or other approaches.

## 1. Introduction

Cancer immunotherapy based on immune-checkpoint blockade has modified cancer treatment paradigms across many tumor types. A major limitation on treatment response to immunotherapy is the inadequate pre-existence of anti-tumor immunity. A number of approaches to initiate de novo immune response have been proposed, with a majority centered on the generation of type I interferons. Of these, agonism of the cGAS/STING pathway is a high priority being pursued for drug development. Here, we discuss the rationale for targeting the STING pathway and comprehensively review the field of STING therapeutics from pre-clinical development through completed clinical trials.

### 1.1. STING and Type I Interferons in Anti-Tumor Immunity

#### Pattern Recognition Receptors

Pattern recognition receptors (PRRs) are germline-encoded proteins that play a central role in immune responses against pathogens as well as recognizing danger molecules associated with endogenous injury [1]. PRRs are activated in response to non-self DNA derived from bacteria or viruses or endogenous self-DNA such as cytosolic DNA derived from tumor cells [2]. There are several families of PRRs, including Toll-like receptors (TLRs), RIG-1 like receptors (RLRs), Nod-like receptors (NLRs), C-type lectin receptors (CLRs), and cytosolic DNA sensors such as the Stimulator of Interferon Genes (STING). The STING protein is a PRR which senses cyclic dinucleotides and induces the expression of type I IFN canonically via dendritic cells [3]. Increasing data suggest that the therapeutic effects of many anticancer modalities, including immunotherapies, depend to a large degree on type I IFN signaling [4,5]. The downstream effects of type I IFNs are mediated through the heterodimeric interferon alpha receptor (IFNαR), which triggers downstream signaling cascades such as JAK-STAT and induces the transcription of IFN-stimulated genes [6]. Type I IFNs exert a variety of effects on different immune cells, including enhancing the cytotoxic ability of natural killer cells and their potential to secrete IFN-γ [7,8], and promoting the differentiation, maturation and migration of antigen-presenting cells [9]. Type I IFN signaling plays a crucial role in innate immune responses, as evidenced by deficient antigen presentation and reduced T cell priming in IFNαR knockout CD8α^+^ dendritic cells [10], and the accelerated metastatic spread of tumor cells in IFNαR knock-out mice [11]. Type I IFNs exert their anti-tumor effects by inhibiting tumor proliferation, and enhancing the expression of MHC class I required for recognition by CD8^+^ T cells. Increasing numbers of publications are suggesting that type I IFNs also inhibit the expression of vascular endothelial growth factor (VEGF) and therefore exert an inhibitory effect on tumor angiogenesis [12,13].

### 1.2. Role of STING in Driving Immune Responses

STING is a transmembrane protein localized to the endoplasmic reticulum which functions as an adaptor protein in the cGAS (cyclic GMP-AMP synthase)-STING pathway [14]. cGAS-STING is a cytosolic DNA-sensing pathway that drives activation of type I IFN and other inflammatory cytokines in the host immune response against tumors [15]. Recognition of cytoplasmic tumor-derived DNA by c-GAS generates cGAMPs which are natural ligands of STING protein. The binding of cGAMP to STING induces transformational changes in STING protein, activating a downstream signaling cascade involving TBK1 and IRF-3, which results in the production of type I IFNs [16].

Early evidence on the role of the STING pathway in immune homeostasis was derived from associations between constitutive STING activation and DNA-mediated inflammatory disorders. Gain-of-function mutations in TMEM173, the gene encoding STING protein, result in constitutive activation of the STING-IFN-β pathway, which is associated with the induction of IFN-response genes such as IP-10. Clinically, the chronic activation of the STING pathway has been linked with an autoinflammatory condition named STING-associated vasculopathy with onset in infancy (SAVI) [17,18]. Symptoms of SAVI usually manifest within the first 8 weeks of life with cutaneous vasculitis, interstitial lung disease and systemic inflammation. Other constitutively active gain-of-function STING mutants have been described as well. A germ-line missense mutation in STING (R284S) was recently described to result in spontaneous STING protein trafficking and downstream signaling in the absence of cGAMP ligand [19]. Another reported mutant STING (G166E) has been associated with the monogenic form of cutaneous lupus. This variant which results from heterozygous mutation in the dimerization domain of the STING causes constitutive type I IFN activation [20].

The type I IFN profile and presence of activated CD8^+^ T cells in the tumor microenvironment have been correlated with favorable outcomes in different solid tumors. Studies using mixed bone marrow chimeras with selective IFN-α/β sensitivity in either innate or adaptive immune compartments have demonstrated that the effect of type I IFNs is most selectively targeted towards innate immune cells, particularly CD8α^+^ dendritic cells [10]. Further experiments in mice deficient in IFN-α/β receptor show that these mice are incapable of rejecting tumors, and that DCs derived from these mice are defective in antigen cross-presentation. These observations provided clues to the involvement of an innate immune pathway in driving T-cell-mediated responses to tumors in vivo. Further mechanistic studies showed that mice deficient in STING protein have defects in CD8^+^ T cell priming and fail to reject immunogenic tumors [16], contrary to mice deficient in other immune adaptors such as TLR4, TLR9 or MyD88 that have a normal CD8^+^ T cell response.

### 1.3. Impact of STING on Tumor Vasculature and Tertiary Lymphoid Structures 

Tumors exhibit genetic instability owing to defects in the expression/functionality of DNA repair proteins [21,22], leading to high intrinsic levels of cytoplasmic DNA and constitutive cGAS/STING activation during tumorigenesis [23,24,25]. To evade immune surveillance mechanisms instigated by the intrinsic activation of STING in tumor cells, many cancers have evolved defects in the STING signaling pathway [25,26,27]. Despite such immune evasion mechanisms (including the genomic or epigenetic silencing of STING expression in tumor cells), genetically unstable cancer cells still enrich the TME with high local concentrations of interstitial dsDNA and/or its cGAS catalyzed product 2’3’ cGAMP capable of activating STING^+^ stromal cells, including dendritic cells and vascular endothelial cells [28,29,30]. STING activation within host stromal cells, particularly endothelial cells, leads to increased vascular perfusion and expression of E-selectin, VCAM-1 and ICAM-1, in association with improved T cell adhesion to the endothelium and the facilitated recruitment of tumor-infiltrating lymphocytes [28,31,32]. Indeed, tumor endothelial STING expression has been tightly correlated with enhanced T cell infiltration and prolonged survival in patients with colon and breast cancer [31]. This paradigm may at least partially explain why cancers with reduced DNA repair proficiency and high comparative mutational burden commonly present as immunologically “hot” tumors that have been reported to be more responsive to interventional immunotherapy [33,34].

However, local stimulation of vascular endothelial cells with prohibitively high concentrations of STING agonists results in cellular apoptosis and vascular necrosis/collapse [35,36,37]. In this light, it is noteworthy that the first-generation STING agonist DMXAA (also known as ASA404 or Vadimezan) was originally developed as a vascular disrupting agent in mice [35]. In early murine tumor models, high-dose (near-MTD) DMXAA-based therapies slowed tumor growth or promoted regression based on vascular necrosis and tumor starvation/hemorrhagic necrosis [36]. However, the tumor microenvironment (TME) remained immunologically sterile/suppressive and treated hosts eventually progressed given a failure to develop durable protective immunity [38,39]. Indeed, high local concentrations of STING agonists have been reported to promote rapid T cell apoptosis [40]. Even when combined with alternate immunotherapies, including tumor antigen-specific CD8^+^ T cell transfer, high-dose DMXAA regimens failed to confer superior therapeutic benefits, owing to contraindicated impact on the recruitment and function of immune cells within the TME [38].

Pioneering work has since led to an appreciation that anti-angiogenic/anti-vascular agents may sponsor tumor vascular normalization when applied using low-dose regimens [41,42]. In contrast to high-dose vascular necrosis-inducing effects, low-dose administration of these drugs promotes the selective pruning away of tumor-associated, immature, dysfunctional vessels and the active fortification of mature, functional blood vessels [41]. Therapeutic vascular normalization therefore results in reduced vascular permeability, interstitial fluid pressure and hypoxia within the TME, in concert with improved tumor blood perfusion and the delivery of systemic treatment agents and immune cells into the TME [31,32,41,42,43]. Remarkably, tumor vascular normalization also results in “immunologic normalization” in the TME, with suppressor/regulatory or pro-angiogenic immune cell subsets (regulatory T cells, myeloid-derived suppressor cells, M2 macrophages) being replaced by pro-inflammatory T cells, M1-macrophages and immunostimulatory antigen-presenting cells [32,41,44]. Recent studies support the ability of low doses of STING agonists cGAMP and ADU-S100 (also known as ML-RR-S2-CDA, MIW815) to promote the local production of anti-angiogenic factors that facilitate vascular normalization when injected intratumorally in mice bearing established breast carcinoma, lung carcinoma or melanoma [31,32]. In keeping with previous reports, the therapeutic efficacy of such intervention was dependent on robust CD8^+^ T cell infiltration of tumors driven primarily by STING activation in, and type I IFN produced by, tumor vascular endothelial cells [31,32,45].

As suggested by recent studies [32], the ability of STING agonist ADU-S100 to promote a state of sustained inflammation within the TME based on conditional changes in vascular activation and the local production of immune recruiting pro-inflammatory cytokines/chemokines also provides fertile soil for the development of tertiary lymphoid structures (TLS). TLS have been classically associated with tissue sites impacted by autoimmune disease or chronic unresolved pathogenic infections [46,47,48]. TLS represent lymph-node-like organizations of immune cells that nucleate around specialized high-endothelial venules (HEV) that conditionally differentiate from CD31^+^ vascular endothelial cells or their progenitors in peripheral tissues under pro-inflammatory conditions [49,50]. Notably, HEV produce a range of type I IFN-inducible chemokines including CCL21 and CXCL13 [51,52], and they express the canonical peripheral node adhesion molecule (PNAd, a ligand for L-selectin/CD62L), facilitating the recruitment of CCR7^+^CD62L^+^ naïve and central memory T cells, CXCR5^+^ B cells and mature CCR7^+^ dendritic cells into TLS [49,50,53]. Within TLS, T cell- and B cell (germinal center)-centric zones develop, within which T cell cross-priming and B cell affinity maturation may occur, resulting in the locoregional differentiation of T effector cells and antibody-producing plasma cells that directly fuel protective anti-tumor/pathogen or deleterious autoimmune host responses [54,55]. Consistent with this paradigm, TLS developed in murine B16 melanomas effectively treated with STING agonist ADU-S100 developed a unique T cell receptor (TCR) repertoire that was not observed in the peripheral T cell population, supporting the operational importance of therapeutic TLS in the melanoma TME. Of major significance, in a series of recent landmark publications, TLS formation in human tumors has been strongly associated with superior patient prognosis and responsiveness to interventional immunotherapies [56,57,58,59,60,61].

Given the reports supporting the ability of low-dose STING agonists to synergize with alternate anti-angiogenic and immune checkpoint strategies in treating established tumors in translational mouse tumor models, it may be anticipated that such combination and/or sustained (low-dose) delivery regimens will optimize tumor-associated vascular normalization, TLS formation within the TME and the evolution of durable protective anti-tumor immunity most capable of effectively controlling the progression of antigenically heterogeneous cancers [31,45,53,62,63].

## 2. STING Agonists in Clinical Development

The substantial pre-clinical anti-tumor activity of STING agonists has led to the development of multiple pharmacologic classes of agents at various stages of being translated into the clinic (Figure 1).

The most prominent tool compound STING agonist broadly used pre-clinically was DMXAA, a vascular disrupting agent known to possess anti-tumor activity. In a randomized phase II clinical trial, feasibility and safety of addition of DMXAA or ASA404 to standard therapy of paclitaxel and carboplatin was assessed in patients with previously untreated advanced non-small cell lung cancer [64]. DMXAA was shown to be well-tolerated with limited evidence of adverse events. However, in a subsequent larger phase III randomized trial assessing the efficacy of paclitaxel and carboplatin with or without ASA404 in patients with advanced NSCLC, no difference was observed in overall survival and progression-free survival between the DMXAA-treated vs. placebo groups, resulting in the termination of the trial during interim analysis due to lack of efficacy [65]. This was contrary to pre-clinical studies using murine models, where intratumoral injection of DMXAA was shown to induce a potent anti-tumor immune response [66,67]. Detailed studies on the function of DMXAA later showed that while DMXAA is a direct ligand for murine STING, polymorphisms in human STING protein leads to the failed binding of DMXAA, rendering it ineffective in humans [68]. Subsequent efforts were directed towards the development of synthetic agonist compounds with improved stability and lower susceptibility to enzymatic degradation that were capable of binding all known alleles of human STING.

### 2.1. Cyclic Dinucleotides

Synthetic cyclic dinucleotides (CDNs) were the first class of STING agonists that entered drug development (Table 1).

*ADU-S100/MIW815* is a synthetic CDN that mimics natural ligands of STING and can activate all known alleles of human STING. Pre-clinical studies using a range of murine tumor models have shown that intratumoral injection of ADU-S100 leads to induction of tumor-specific CD8^+^ T cells [69,70], and that addition of ADU-S100 to anti-PD1 or anti-PD1/anti-CTLA4 antibodies in combination therapies results in enhanced tumor-specific T cell responses and superior anti-tumor efficacy [71,72]. ADU-S100/MIW815 was studied clinically in three clinical trials (NCT03172936, NCT02675439, NCT03937141). In a phase I dose escalation clinical trial, the safety of intratumoral ADU-S100/MIW815 monotherapy (days 1,8 and 15 in a 28-day cycle) was assessed in patients with advanced/metastatic solid tumors or lymphomas (NCT02675439). In this trial involving 40 patients, no dose-limiting toxicities were reported. The most common reported adverse events included pyrexia, pain at the site of injection and headache. Intratumoral injection of ADU-S100/MIW815 was associated with increased levels of inflammatory cytokines such as IL-6, IFN-β1 and MCP1/CCL2. Preliminary results showed partial responses in two patients with Merkel cell carcinoma and parotid gland carcinoma, both of whom had previously received anti-PD1 therapy. In addition, 11 out of 40 patients achieved stable disease. The preliminary results of the phase Ib study on ADU-S100/MIW815 in combination with anti-PD1 spartalizumab in patients with advanced/metastatic solid tumors or lymphoma was presented at ASCO 2019 [73]. In this dose escalation study, patients received intratumoral injections of ADU-S100 either weekly or every 4 weeks in combination with intravenous spartalizumab. No dose-limiting toxicities were reported, with the most common treatment related adverse events including pain at the site of injection, pyrexia and diarrhea. Increased levels of liver enzymes were reported as grade 3/4 treatment related adverse events (TRAEs) in 3% of patients. Partial responses were observed in a number of patients with anti-PD1/L1 naïve triple negative breast cancer and anti-PD1 relapsed/refractory melanomas. In the other two ongoing trials, ADU-S100 was clinically tested in combination with pembrolizumab in patients with PD-L1 positive recurrent or metastatic head and neck squamous cell carcinoma (HNSCC) (NCT03937141), and in combination with ipilimumab in patients with advanced/metastatic solid tumors or lymphomas (NCT02675439). Data for these studies has not yet been released, however, these clinical trials have been reported as discontinued.

*MK-1454* is synthetic CDN that has advanced into clinical development. The preliminary data of a phase I, open-label, multi-arm first-in-human study on MK-1454 was first publicly released at ESMO 2018 [74]. In this dose escalation study, patients with advanced solid tumors or lymphomas received intratumoral injections of MK-1454 either as monotherapy or in combination with anti-PD1 agent pembrolizumab (NCT03010176). Although no complete or partial responses were observed in the monotherapy arm, 24% of patients (6/25) in the combination therapy arm demonstrated partial responses which were durable for more than 6 months, with 83% median reduction in size of both injected and non-injected lesions. TRAEs were reported in 83% of patients in the monotherapy arm and 82% of patients in the combination therapy arm, of which 9% and 14% were grade 3 or higher adverse events respectively, resulting in discontinuation of treatment in 7% of patients in the combination therapy arm. Reported TRAEs included pain at the injection site, pyrexia, fatigue and pruritus. In patients receiving MK-1454, dose-dependent increases were observed in serum IP-10, STING-induced gene signature, and IL-6 (in smaller subset of patients) in both responder and non-responder groups. Intratumoral MK-1454 is also being tested in patients with metastatic or unresectable recurrent HNSCC as single agent or in combination with pembrolizumab (NCT04220866).

To circumvent the limitations of intratumoral delivery, recent efforts have focused on developing STING agonists with stable physical properties that could be delivered systemically. Although multiple compounds are being developed for systemic delivery in preclinical murine models, only a few have entered clinical testing. *SB11285* is a small molecule CDN STING agonist which is being clinically evaluated for intravenous administration in patients with advanced solid tumors. Pre-clinical data using SB11285 in syngeneic mouse models showed significantly higher inhibition of tumor growth in mice injected with intratumoral SB11285 compared to the control group. Additionally, SB11285 in combination with cyclophosphamide resulted in a significant synergistic anti-tumor effect [75]. The ongoing phase 1a/1b non-randomized, dose escalation study on SB11285 aims to evaluate the efficacy of intravenous SB11285 either as a single agent or in combination with Atezolizumab in patients with advanced solid tumors (NCT04096638). After determination of dose limiting toxicities and maximum tolerated dose of SB11285, this study aims to assess the anti-tumor activity of IV SB11285 in combination with atezolizumab.

Pre-clinical evaluation of a novel STING agonist *BMS-986301* in CT26 and MC38 murine tumor models yielded promising results, with >90% complete regression seen in injected and non-injected tumors as opposed to 13% with ADU-S100. In the CT26 model, a single dose of BMS-986301 combined with anti-PD1 resulted complete regression of 80% of injected and non-injected tumors when no regressions were seen with anti-PD1 alone [76]. BMS-986301 is currently in clinical testing as intratumoral or intramuscular injection alone or in combination with nivolumab and ipilimumab in patients with advanced solid cancers that have not responded to checkpoint inhibitor therapy (NCT03956680). In subcutaneous syngeneic murine tumor models, a single intratumoral injection of *BI-STING*, a CDN mimicking natural STING ligand, resulted in dose-dependent anti-tumor activity and inhibition of tumor development upon re-challenge [77]. BI 1387446 is one of the BI-STING compounds that has entered clinical testing. In an ongoing phase I first in human trial, the maximum tolerated dose and tolerability of a single intratumoral injection of BI 1387446 alone or in combination with BI 754091 (an anti-PD1 monoclonal antibody) is being assessed in patients with advanced, unresectable and/or metastatic solid tumors (NCT04147234).

### 2.2. Non-Cyclic Dinucleotides

*MK-2118* is a STING agonist of unreported structure which is being tested as an intratumoral or subcutaneous injection alone or in combination with pembrolizumab in patients with advanced solid tumors or lymphomas (NCT03249792). In an early dosing phase 1 clinical trial, *GSK3745417*, a non-CDN small molecule with dimeric amidobenzimidazole (ABZI) scaffold, is being tested as an intravenous injection either alone or in combination with pembrolizumab in patients with refractory/relapsed solid tumors (NCT03843359). *SNX281* is a novel small molecule therapeutic developed as a STING agonist, which is active against all isoforms of human STING and has stable drug properties thus permitting systemic delivery. In pre-clinical models, a single intravenous dose of SNX281 in mice bearing CT26 colorectal tumors resulted in complete regression of tumors. SNX281 also synergized with anti-PD1 agents in inhibiting tumor growth and improving the overall survival of tumor-bearing mice. In an ongoing phase 1 clinical trial, the safety, tolerability and maximum tolerated dose of systemic SNX281 will be assessed in patients with advanced solid tumors and lymphomas. This trial is comprised of two treatment arms, in which intravenous SNX281 is given either as monotherapy or in combination with pembrolizumab, in a dose escalation followed by dose expansion phase to determine the recommended dose for phase 2 studies (NCT04609579). *TAK-676* is another small molecule STING agonist with an undisclosed structure that is now under clinical investigation in a phase I dose escalation study. This trial aims to determine the safety and tolerability of intravenous TAK-676 as monotherapy and in combination with pembrolizumab in patients with advanced or metastatic solid tumors (NCT04420884).

*E7766* belongs to the novel class of macrocycle-bridged STING agonists (MBSAs). MBSAs show superior in vitro activity against all major human STING genotypes. E7766 binds to both human and mouse STING protein and show activity against a broader range of human STING genotypes compared to reference CDNs. A single intratumoral injection of E7766 was shown to significantly reduce the growth of subcutaneous tumors in mice [78]. In mice bearing CT26 tumors in both subcutaneous tissue and liver, a single intratumoral injection of E776 led to resolution of tumors in 90% of treated mice with no recurrence over 8 months [79]. Intravesical administration of E7766 in mice with BCG-unresponsive non-muscle invasive bladder cancer was associated with robust IFN-β gene induction and a dose-dependent anti-tumor response [80]. The clinical efficacy of intratumoral injection of E7766 is being evaluated in a phase 1/1b clinical trial as a monotherapy in patients with advanced solid tumors or lymphomas (NCT04144140).

### 2.3. Bacterial Vectors

Bacterial vectors have been used as a novel approach for delivering STING agonists to intratumoral antigen-presenting cells. One such vector is *SYNB1891*, a non-pathogenic *E. Coli Nissle* strain engineered to express cyclic di-AMP-producing enzymes in response to the hypoxic environment of tumor, while remaining localized to the tumor microenvironment. Pre-clinical studies show that intratumoral injection of SYNB1891 to B16.F10 melanoma tumor-bearing mice induces production of type I IFNs and leads to significant reduction in tumor growth eight days after treatment [81]. In an ongoing phase I clinical trial, anti-tumor efficacy of intratumoral SYNB1891 as monotherapy or in combination with Atezolizumab is being evaluated in patients with advanced/metastatic solid tumors and lymphoma (NCT04167137). Another novel bacterial-based immunotherapy is *STACT*, an attenuated Salmonella Typhimurium strain that carries an inhibitory microRNA to TREX-1. TREX-1 exonuclease prevents activation of STING pathway by degrading cytosolic DNA. Pre-clinical studies show that intravenously delivered STACT-TREX-1 specifically colonizes to the myeloid compartment of the tumor microenvironment [82]. In CT26 and MC38 murine models, intravenous injection of STACT-TREX-1 was associated with very low systemic levels of inflammatory cytokines, demonstrating tumor specific colonization of STACT-TREX-1, tumor regression and durable immunity upon re-challenge [83]. STACT-TREX-1 is advantageous over early-generation STING agonists as it can be administrated systemically and thus can target a wider range of tumors. This platform is yet to enter clinical testing.

## 3. STING Agonists in Pre-Clinical Evaluations

The chemical structure and stability of many clinically developed STING agonists have limited their use as systemic immunotherapeutics. In the past few years, novel cyclic CDN and non-CDN STING agonist compounds with improved structural properties, higher potency and affinity towards human STING, and with potential for systemic delivery have been identified and shown promising results in pre-clinical studies (Table 2).

### 3.1. Novel Cyclic Dinucleotides

*JNJ-67544412 (JNJ-4412)* is a recently developed CDN STING agonist, which is claimed to bind to all major alleles of human STING with a stronger affinity than other known CDNs. In syngeneic mouse tumor models, intratumoral injection of JNJ-4412 was shown to result in increased levels of pro-inflammatory cytokines in tumor and plasma, increased frequencies of CD8^+^ T cells, loss of vascularization, increased apoptosis and significant regression of both injected and contralateral tumors. Transient body weight loss was reported as an adverse event but was alleviated with less frequent dosing [84]. In a recent study, *3′3′-cyclic AIMP* developed as a CDN STING agonist was tested in a murine model of hepatocellular carcinoma (HCC). Intraperitoneal injection of 3′3′-cyclic AIMP to STING-deficient mice with HCC resulted in a reduced tumor burden exemplified by smaller liver surface nodules, an increased number of CD8^+^ T cells and increased apoptotic cell death within the tumor. When 3’3’-cyclic AIMP was administered at a later stage after HCC development, tumor regression was observed in the majority of tumors; however, new tumors developed that were not responsive to CDN treatment, supporting the efficacy of this agent in reducing tumor burden but its inability to completely resolve all lesions in multi-focal disease models [106]. Another novel CDN, *GSK532*, has shown improved stability with minimal degradation in human whole blood, and is able to induce cytokine responses in human cells with different STING haplotypes. In addition, intratumoral injection of GSK532 to mice harboring CT26 tumor cells was shown to induce anti-tumor responses in both injected and uninjected tumors [86]. This compound is yet to enter clinical testing.

### 3.2. Next-Generation Non-Cyclic Dinucleotides

*TTI-10001* is a non-CDN small molecule STING agonist which is able to bind to all five human STING alleles as well as to the murine STING protein. Intratumoral administration of TTI-10001 in murine models of syngeneic tumors was found to be safe and associated with increased levels of phospho-IRF3, pro-inflammatory cytokines, and anti-tumor activity [90]. *Selvita agonists* are a group of recently developed small-molecule, non-nucleotide, non-macrocyclic STING agonists. These agonists selectively bind to both mouse and human STING proteins and have tunable properties with enhanced plasma stability and permeability, making them potential candidates for systemic delivery [89]. In vitro studies on peripheral blood mononuclear cells and the THP1 monocytic cell line show that *Selvita* agonists induce the expression of inflammatory cytokines and upregulate maturation markers on the surface of APCs. A group of selective non-CDN non-macrocylic small molecule compounds (*Ryvu’s agonists*) have shown promising results in pre-clinical animal models [87]. These agonists bind to STING proteins of different species and are able to induce DC maturation and cytokine expression from human PBMCs irrespective of the STING haplotype. Systemic administration of these compounds (route not specified) in mice bearing CT26 colorectal cancer cells resulted in complete tumor regression and the development of immunological memory [87].

*ALG-031048* is a novel STING agonist that has shown higher stability in in vitro studies compared to natural STING ligand and STING agonist ADU-S100. Intratumoral injection of ALG-031048 in mice bearing CT26 tumor cells resulted in tumor regression in 90% of mice (compared to 44% with ADU-S100). Treated mice were resistant to tumor development after re-challenge with the same tumor cell line [95]. In another study, intratumoral injection of ALG-031048 to mice bearing Hepa1–6 hepatocellular carcinoma tumor cells resulted in a mean tumor regression of 88% compared to 72.4% regression after treatment with anti-PD1 antibody. Treatment with ALG-031048 was associated with a dose-dependent increase in the level of cytokines such as IFN-β1, IFN-γ, TNF-α, IL-6, MIP-1α and MCP-1. The anti-tumor efficacy of ALG-031048 was also shown after subcutaneous injection, and the combination of subcutaneous ALG-031048 with anti-PDL-1 agent, atezolizumab, further enhanced tumor growth inhibition from 60% with atezolizumab alone to 77% with combination therapy [107]. *MSA-1* STING agonist compound has also shown robust anti-tumor efficacy when injected intratumorally to mice bearing MC38 syngeneic colon carcinomas. Complete responses were observed in 100% of tumors in mice receiving the highest tolerated dose of intratumoral MSA-1. Combination of MSA-1 with anti-PD1 antibody (mDX400) resulted in the restoration of T cell responses in anti-PD1 unresponsive tumors, further supporting the synergic anti-tumor activity of STING agonists with anti-PD1 therapy [94]. *CRD-5500*, a next-generation small molecule STING agonist, has shown to be effective via intratumoral and systemic routes and also as an antibody-drug conjugate with Trastuzumab. The fact that CRD-5500 can be dosed via multiple routes makes it a favorable agent for future clinical development. Pre-clinical data show that both intravenous and intratumoral injections of CRD-5500 induce tumor regression in murine CT26 colon carcinoma models engineered to express human STING, and this anti-tumor effect is further amplified when CRD-5500 is combined with check point inhibitor therapy [92].

### 3.3. ENPP1 Inhibitors

*ENPP1* is a transmembrane phosphodiesterase known for its central role in purinergic signaling [108]. Recent studies show that ENPP1 can downregulate cGAS-STING signaling by hydrolyzing cGAMP, the natural STING ligand [109]. This finding mainly stems from observations that cGAMP derived from ENPP1 knock-out cells has a higher ability to activate STING, and the in vitro inhibition of ENPP1 amplifies cGAS-STING signaling [110]. Orally available small molecules that inhibit ENPP1 have been developed as novel STING agonists [98]. In one such study, small molecule compounds with strong binding affinity towards ENPP1 were identified using computational methods and direct binding assays. These compounds were shown to bind specifically to ENPP1 vs. other members of the ENPP family and to activate the STING pathway [98]. *MV-626* is a selective ENPP1 inhibitor with 100% bioavailability that has been studied in pre-clinical models. Intraperitoneal injection of MV-626 alone or in combination with radiation therapy in mice implanted with Panc02-SIY pancreatic adenocarcinoma tumors resulted in a durable anti-tumor immune response and improved overall survival [99]. *SR-8314* and *SR-8291* are both highly selective ENPP1 inhibitors that have shown promising in vivo efficacy in syngeneic murine tumor models. Intraperitoneal injection of *SR-8314* and *SR-8291* led to increased frequencies of CD4^+^ and CD8^+^ T cells and a decrease in tumor-associated macrophages in tumor-bearing mice [97]. *SR8541A* is another small molecule ENPP1 inhibitor which in recent in vitro studies has been shown to stimulate the migration and infiltration of peripheral blood myeloid cells into the tumor microenvironment [96]. Though the preclinical efficacy of these orally bioavailable compounds has been promising after intraperitoneal delivery, they are yet to be explored in models when delivered orally.

### 3.4. Novel STING Agonists for Systemic Delivery

Most CDN-based STING agonists in clinical development are delivered intratumorally due to their poor stability, which makes their utility limited to accessible tumors. To circumvent this, recent efforts have focused on the development of compounds that are capable of activating STING and are structurally stable for systemic delivery. A class of novel small molecule non-cyclic dinucleotides intended for systemic delivery are *amidobenzimidazole (ABZI)-based* compounds. One such compound, called *Compound 3*, was shown to activate STING and induce IFN-β production approximately 400-fold stronger than natural cGAMP. Intravenous administration of Compound 3, an ABZI-based compound, in the CT26 mouse model of colorectal cancer led to significant inhibition in tumor growth and improved overall survival, effects that were dependent on CD8^+^ T cell-mediated immune responses [111].

In a recent study, *SR-717* was identified as a novel non-nucleotide cGAMP mimetic with potential for systemic delivery. Daily intraperitoneal injections of SR-717 to mice bearing established B16.F10 melanomas for one week resulted in a significant reduction in tumor growth and the increased survival of tumor-bearing mice [112]. Additionally, intraperitoneally administered SR-717 was shown to inhibit the formation of pulmonary nodules in a murine model of pulmonary metastasis. *MSA-2* is another novel and orally available STING agonist with distinct structural properties that makes it stable for systemic delivery [113]. In vitro assays show that MSA-2 has a higher cellular permeability compared to other CDNs. A distinct property of MSA-2 is its binding mode which induces a closed conformation of STING, similar to the natural cGAMP ligand. Lower extracellular pH, as seen in the acidic microenvironment of tumors, was also associated with a higher intracellular concentration of MSA-2 and greater cellular potency. Using various syngeneic murine systems including MC38 and CT26 colorectal carcinoma, B16F10 melanoma and LL-2 lung cancer models, the combination of subcutaneous or orally administered MSA-2 with intraperitoneal anti-PD1 was found to have a synergistic effect in inhibiting tumor growth and improving the overall survival of tumor-bearing mice vs. component monotherapies [113].

*JNJ-‘6196* has been developed as a next-generation STING agonist, which binds to STING protein with weaker affinity and a faster off rate, but is able to activate dendritic cells and induce cytokine expression with a higher potency compared to other CDNs. JNJ-‘6196 can be administered systemically via the intravenous route, where it has been shown to effectively eliminate bilateral tumors, promote immune-mediated resistance to tumor re-challenge, and to improve the efficacy of checkpoint inhibitors in PD-1 non-responsive tumor models in mice [91]. Further investigation into the functional properties of JNJ-‘6196 showed that its systemic efficacy is related to the intensity of cytokine gene induction rather than the induced gene signature [91]. The functional efficacy of JNJ-‘6196 provided via the intravenous route as well as its ability to synergize with checkpoint inhibitors make this compound an intriguing candidate for clinical development.

## 4. Novel STING Agonist Delivery Platforms

Antibody drug conjugates (ADCs) represent a novel approach for systemic yet targeted STING agonist delivery. STING antibody-drug conjugates have shown promising pre-clinical results [100]. In in vitro assays, STING agonist ADCs showed a 100-fold higher potency in inducing inflammatory cytokines and augmenting PBMC-mediated cancer cell death compared to free STING agonist. A single intravenous injection of targeted STING ADC in two murine tumor models led to significant inhibition in tumor growth when compared to systemically injected diABZI [100]. Interestingly, this superior anti-tumor effect was associated with significantly lower levels of systemic cytokines, supporting the ability of STING ADC to promote an effective anti-tumor response locally. *XMT-2056* is the first STING ADC developed through the Immunosynthen platform that is expected to enter phase I clinical testing in 2022.

Nanoparticle (NP) vaccines or nanovaccines are a novel delivery platform that optimizes the spatiotemporal coordination of antigen presentation to APCs with innate immune responses in order to amplify the cytotoxic T cell responses [114]. *PC7A NP* is a synthetic nanoparticle which is shown to improve the delivery and cross-presentation of antigens by APCs, activate type I IFN genes, and stimulate specific CTL responses in mice [115]. Of note, this CTL response is independent of the toll-like receptor or MAVS pathways, but rather dependent on the activation of the cGAS/STING pathway. PC7A vaccination inhibited tumor growth in B16-F10 melanoma and MC38 colon carcinoma murine models. In addition, PC7A was shown to synergize with PD-1 inhibitor therapy, and to confer resistance against tumor development upon re-challenge, suggesting the formation of protective memory responses [115]. A study on the efficacy of liposomal nanoparticles delivering cGAMP in mice bearing basal-like triple negative breast cancer cells showed that intravenously delivered cGAMP-NPs accumulate within macrophages at the site of the tumor, skew them from M2-to-M1 inflammatory phenotype, and enhance their expression of MHC II and co-stimulatory molecules. In addition, in TNBC and B16F10 melanoma murine models with poor intrinsic response to anti-PD1 monotherapy, cGAMP-NP injection was shown to result in a more effective and durable anti-tumor response compared to soluble cGAMP [105]. *ONM-500* nanovaccine was shown to effectively accumulate in lymph nodes after subcutaneous injection in mice [101]. Administration of ONM-500 containing full length E6/E7 proteins to mice bearing TC-1 cervical cancer cells resulted in a significant improvement in animal survival and was associated with long-term anti-tumor response in re-challenge studies. In a recent study, a novel nanovaccine with redox-responsive neoantigen-polymer conjugate and STING agonist DMXAA was developed [102]. Subcutaneous injection of this formulated nanovaccine to mice bearing B16-F10 melanoma cells in combination with anti-PD1 was found to significantly improve survival compared to non-formulated neoantigen peptides and was associated with higher systemic levels of inflammatory cytokines [102].

Antigen-presenting cell (APC)-targeted tumor vaccines are another recent development in the field of cancer therapeutics. One such formulation combining granulocyte macrophage colony stimulating factor (GCSF)-secreting pancreatic tumor cells with attenuated Listeria-expressing mesothelin has shown promising results in patients with pancreatic carcinoma [116]. A novel STING-pathway-targeting vaccine based on this platform is STINGVAX, in which synthetic CDNs are used as adjuvants and co-formulated with irradiated GCSF-expressing vaccine cells [117]. In a recent study, subcutaneous injection of STINGVAX to mice bearing B16 tumor cells was associated with STING-dependent IRF3 and type I IFN expression, and tumor regression. This anti-tumor response was amplified when STINGVAX was combined with PD-1 blocking antibody. Similar results were observed with intraperitoneal injection of STINGVAX in combination with anti-PD1 therapy in the CT26 murine model [117].

Exosomes are cell-derived nanovesicles that transmit signals and molecules between cells. In recent years, there has been an emerging interest in exosome-based therapies as a platform for delivering anti-tumor agents. In animal models, vaccination with tumor-asociated exosome-loaded dendritic cells or T cells has been shown to enhance immune responses against tumors [118,119]. Multiple groups have recently focused on developing exosome-based therapeutics for the delivery of STING agonists to tumors. exoSTING is an engineered exosome which expresses high levels of an exosomal surface glycoprotein called Protein X and is loaded with a STING agonist. Intratumoral injection of exoSTING to mice bearing checkpoint refractory melanoma tumor cells results in a more potent tumor-specific immune responses and lower systemic cytokine production levels when compared to free STING administration. Combination treatment using exoSTING with PD1 checkpoint inhibitor has also been shown to further enhance anti-tumor immune responses [103].

## 5. Conclusions and Future Directions

Since the initial discovery of STING and the potency of agonizing the pathway in the pre-clinical setting, there have been accelerated attempts to develop STING agonists as effective immunotherapeutic agents. The presence of a human autoimmune phenotype (SAVI) and an ever-expanding number of pre-clinical and clinical studies support the potential for numerous STING agonist compounds in boosting anti-tumor immunity and enhancing the effects of existing immunotherapies. Despite this, there remain substantial barriers for the clinical targeting of the cGAS-STING pathway. The first-generation CDN STING agonists to enter clinical testing were administered intratumorally. This presents challenges as the approach limits the use of STING agonists to only accessible tumors. Numerous efforts are under investigation towards the development of compounds with improved properties enabling their stability for systemic delivery, with promising results obtained in pre-clinical studies implementing these compounds. Advances in imaging-guided drug delivery methods are also expected to enhance the applicability of intratumorally administered STING agonists to a wider range of tumor types. A concern raised over the systemic administration of STING agonists is whether these agents induce a state of pathologic inflammation, as the overactivation of STING has been implicated in a broad range of autoimmune conditions. The off-target effect of STING activation on other immune cells such as effector T cells could be another area of concern. In in vitro studies, STING activation was associated with the induction of stress and apoptosis in T cells [120]. Whether this would dampen T-cell-mediated responses in the clinical setting and how such effects would impact the development of memory responses need to be further explored. Few studies have suggested that certain regulatory factors are induced in response to cGAS-STING pathway activation, which could potentially attenuate the anti-tumor efficacy of STING agonists and lead to treatment resistance. A rapid increase in the expression of PD-L1, COX2 and IDO in the tumor microenvironment was observed after intratumoral injection of a STING agonist [121]. Using digital spatial profiling, it was also recently shown that after intratumoral injection of STING-activating nanoparticles, the expression of B7-H3 and S100A9 immune checkpoints, which are associated with tumor immune evasion is increased. Notably, this effect was partially abolished when STING-NP was used in combination with an immune checkpoint inhibitor [122]. Whether activation of the cGAS-STING pathway could elicit negative feedback loops that would dampen the effect of STING agonists is not known, and further studies are justified to determine potential mechanisms of resistance to STING agonist-based therapy. An additional limitation to the effectiveness of STING agonist therapies is the selective silencing of the cGAS-STING pathway in certain types of tumors. Indeed, the suppression of STING signaling as a result of the epigenetic silencing of promoter regions or loss-of-function mutations have been reported in tumors [123]. An improved understanding of the mechanisms underlying STING signaling defects in tumors and corrective protocols that would augment the response to STING agonists in the tumor microenvironment for therapeutic benefit is clearly warranted.

## Figures and Tables

**Figure 1 cancers-13-02695-f001:**
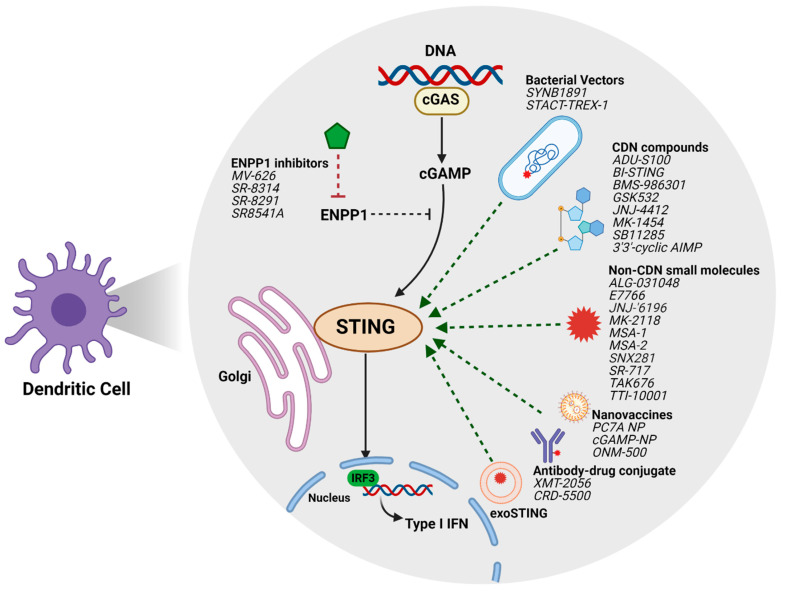
Novel STING agonist strategies and agents in development.

**Table 1 cancers-13-02695-t001:** STING agonists in clinical development. HNSCC: head and neck squamous cell carcinoma, IM: intramuscular, IT: intratumoral, IV: intravenous, SQ: subcutaneous.

Agent		Route of Delivery	Phase	Type of Cancer	Clinical Trial NCT Code
ADU-S100/MIW815	Single agent or + Ipilimumab	IT	Phase I	Advanced/Metastatic Solid Tumors or Lymphomas	NCT02675439
	+ Pembrolizumab	IT	Phase II	PD-L1 positive recurrent or metastatic HNSCC	NCT03937141
	+ PDR001	IT	Phase Ib	Advanced/Metastatic Solid Tumors or Lymphomas	NCT03172936
MK-1454	Single agent or + Pembrolizumab	IT	Phase I	Advanced/Metastatic Solid Tumors or Lymphomas	NCT03010176
	+ Pembrolizumab	IT	Phase II	Metastatic or Unresectable, Recurrent HNSCC	NCT04220866
MK-2118	+ Pembrolizumab	IT/SQ	Phase I	Advanced/Metastatic Solid Tumors or Lymphomas	NCT03249792
SB11285	Single agent or + Atezolizumab	IV	Phase Ia/Ib	Advanced Solid Tumors	NCT04096638
GSK3745417	Single agent or + Pembrolizumab	IV	Phase I	Advanced Solid Tumors	NCT03843359
BMS-986301	Single agent or + Nivolumab/ Ipilimumab	IT/IM	Phase I	Advanced Solid Tumors	NCT03956680
BI-STING (BI 1387446)	Single agent or + BI 754091 (anti-PD1 monoclonal antibody)	IT	Phase I	Advanced Solid Tumors	NCT04147234
E7766	Single agent	IT	Phase I/Ib	Advanced Solid Tumors or Lymphomas	NCT04144140
TAK-676	Single agent or + Pembrolizumab	IV	Phase I	Advanced Solid Tumors	NCT04420884
SNX281	Single agent or + Pembrolizumab	IV	Phase I	Advanced Solid Tumors or Lymphomas	NCT04609579
SYNB1891	Single agent or + Atezolizumab	IT	Phase I	Advanced Solid Tumors or Lymphomas	NCT04167137

**Table 2 cancers-13-02695-t002:** STING-targeting compounds in pre-clinical stage.

Agent	Structure/Properties	Route of Delivery	Tumor Model	Findings	References
**Cyclic dinucleotide (CDN)**					
JNJ-67544412 (JNJ-4412)	Cyclic dinucleotide, Potently binds to all major human STING alleles	Intratumoral	Subcutaneous syngeneic murine tumor models	- Tumor regression, induction of proinflammatory cytokines such as IFN-α, IFN-β, IP-10, TNF-α, IL-6 and MCP-1 in tumor and plasma, inhibition in growth of contralateral tumors. - Enhanced dose-dependent efficacy when combined with anti-PD1.	[84]
BI-STING	Mimics natural STING ligand	Intratumoral	Subcutaneous syngeneic murine tumor models	- Single dose of intratumoral BI-STING results in transient increase in cytokine levels, dose-dependent local tumor control. No tumor developed upon re-challenge. - Tumor control improved when combined with anti-PD1 - ELISPOT: higher number of immunospots in splenocytes from BI-STING-treated animals showing induction of tumor specific immune response.	[77]
3′3′-cyclic 3′3′-cAIMP	Cyclic dinucleotide	Not specified	Mouse model of mutagen-induced hepatocellular carcinoma	- Treatment of mice after HCC development efficiently reduced tumor size. - Initiation of treatment at later stage of disease development resulted in regression of the majority of tumors, but new treatment-unresponsive tumors were detected.	[85]
GSK532	Cyclic dinucleotide	Intratumoral	CT26 murine syngeneic model	- Strong anti-tumor effect in both the injected and uninjected tumors. - Cured mice were resistant to re-challenge with the same tumor cell line.	[86]
**Non-CDN Agonists**					
Ryvu’s agonists	Selective non-nucleotide, non-macrocyclic, small molecule compounds, potential for systemic administration	Not specified	CT26 murine syngeneic model	- Dose-dependent upregulation of STING-dependent pro-inflammatory cytokines. - Complete tumor remission and development of immunological memory against cancer cells.	[87]
GF3-002	Novel low-molecular-weight organic molecule, not based on a CDN	In vitro	In vitro assays	- Confirmed binding to WT STING and production of IFN-β after treatment of dendritic cells with GF3-002.	[88]
Selvita agonists	Selective non-nucleotide, non-macrocyclic, small molecule compounds, structurally unrelated to known CDNs, tunable properties with enhanced plasma stability and permeability, potential for systemic administration	In vitro	In vitro assays	- Induction of cytokine responses (IFN-β, TNF-α) in human PBMC, human monocyte derived macrophage, and human DCs with various STING haplotypes including refractory alleles. - Induction of pro-inflammatory cytokine profile and up-regulation of the maturation markers on human APCs.	[89]
TTI-10001	Non-CDN small molecule STING agonist	Intratumoral	Multiple syngeneic murine tumor models	- Well-tolerated in vivo; results in increased expression of pro-inflammatory cytokines, and anti-tumor activity.	[90]
JNJ-‘6196	Next-generation STING agonist; binds to STING with weaker affinity and a faster off rate, but more potent than other CDNs in activating dendritic cells	Intravenous	Murine tumor models (not specified)	- Eliminates bilateral tumors, and provides immunity to further re-challenge. - Increases the effectiveness of checkpoint inhibitors, turning a PD-1 resistant model into a responsive model.	[91]
CRD5500	Next-generation small molecule STING agonist. Activates all five common human STING variants. Delivery via different routes (IV or SC) or as an antibody drug conjugate	Intravenous, subcutaneous, Antibody-drug conjugate (ADC) with Trastuzumab	CT26 syngeneic murine model	- In vitro: causes maturation of hDCs and the release of innate and adaptive inflammatory cytokines such as IFN-β and TNF-α. - In vivo administration (IT or systemically): tumor regression in CT26 syngeneic tumors containing human STING.	[92]
CS-1018, CS-1020 and CS-1010	STING agonists with higher potency in activating mouse and human STING variants than natural ligand cGAMP	Intratumoral	B16F10 and MC38 murine tumor models	- All compounds showed dose-dependent anti-tumor activity in MC38 or B16F10 syngeneic models. - Tumor-free treated mice developed tumor specific immunologic memory in MC38 murine model.	[93]
MSA-1	Novel STING agonist with higher potency in activating STING protein than cGAMP	Intratumoral	MC38 syngeneic tumors, CT26 and B16-F10 tumor models	- Complete responses observed in 100% of MC38 tumors. - Restoration of T cell responses (in serum and tumors) of mice with anti-PD1 unresponsive tumors when combined with anti-PD1.	[94]
ALG-031048	Novel STING agonist with high potency and superior stability	Intratumoral, Subcutaneous	Syngeneic CT26 colorectal, B16F10 melanoma, and Hepa1–6 HCC models	- Tumor regression in 90% of mice bearing CT26 tumors (vs. 44% with ADU-S100). Treated mice were resistant to tumor development after re-challenge. - Mean tumor regression of 88% in HCC tumor-bearing mice vs. 72.4% regression with anti-PD1 antibody. - Dose-dependent increase in plasma levels of IFN-β1, IFN-γ, TNF-α, IL-6, MIP-1α and MCP-1 - Subcutaneous ALG-031048 improved anti-tumor efficacy of anti-PDL-1 agent, atezolizumab.	[95]
**Macrocyclic STING Agonist**					
E7766	Macrocyclic STING agonist with superior in vitro activity against all major human STING genotypes, chemical and metabolic stability, conferred by conformational rigidity of the unique macrocycle bridge	Intravesical, Intratumoral	Murine anti-PD1 insensitive NMIBC tumor models, subcutaneous tumor models	- Intravesical: dose-dependent anti-tumor effect vs. anti-PD1 which was ineffective. Tumor-free animals rejected re-challenge of same tumor cell line. Activation of IFN pathway, T cell infiltration, NK activity, induction of IFN-β and CXCL10 inside the bladder cavity and in the urine. - Intratumoral: single IT injection led to complete regression or significant tumor growth delay.	[78,80]
**ENPP1 Inhibitor**					
SR-8541A	Small molecule ENPP1 inhibitor	In vitro	In vitro assays	- Stimulates the migration and infiltration of immune cells (PBMC) into cancer spheroids, increases expression of IFN-β, ISG15 and CXCL10. - ENPP1 CRISPR knockout cell models confirmed that the drug effect is ENPP1-dependent.	[96]
SR-8314	Analog of SR-8291 (a highly selective ENPP1 inhibitor)	Intraperitoneal	Syngeneic murine tumor model	- Increase in gene expression of IFN-β, ISG15 and CXCL10 and secretion of IFN-β in SR-8314-treated THP1 cells. - Anti-tumor activity, increase in CD3^+^, CD4^+^ and CD8^+^ T cells in both SR-8314 and SR-8291-treated tumors, decrease in tumor-associated macrophages in SR-8314-treated tumors.	[97]
Orally available ENPP1 inhibitors	Small molecule compounds with strong binding affinity towards ENPP1	In vitro	In vitro assays	- Specific and high binding affinity to ENPP1 with no effect on other members of the ENPP family, activation of STING pathway. - One of lead compounds is currently under investigation for ADME-Tox, PK and efficacy.	[98]
MV-626	Selective ENPP1 inhibitor with 100% oral bioavailability	Intraperitoneal	Panc02-SIY and MC38 murine tumor models	- Therapeutic doses were well tolerated in mice, without toxicity or clinically significant increases in systemic cytokine levels. - Systemic MV- 626 monotherapy caused tumor growth delay. MV-626 plus radiation therapy significantly increased overall survival.	[99]
**Novel Delivery Systems**					
Antibody drug conjugates (ADC)	STING agonist ADCs	Intravenous	Multiple xenograft and syngeneic murine models	- 100-fold more potency in inducing inflammatory cytokine expression compared to free agonist. - Inflammatory cytokines were tumor localized while systemic levels remained low. - Single IV injection of targeted STING ADC in tumor-bearing mice significantly inhibited tumor growth compared to systemically injected diABZI.	[100]
ONM-500 nanovaccine	Novel pH-sensitive polymer that forms an antigen-encapsulating nanoparticle and functions both as a carrier for antigen delivery to DCs and as an adjuvant, activating the STING pathway	Subcutaneous	TC-1 cervical cancer murine model	- Effective binding to human STING protein. - Effective delivery of antigens in vivo to LNs to elicit an antigen-specific CTL response. - ONM-500 nanovaccine containing full-length E6/E7 protein resulted in 100% overall survival of TC-1 bearing mice at 55 days. - Long-term antigen-specific anti-tumor memory response in re-challenge study.	[101]
Neoantigen nanovaccine	Redox-responsive neoantigen-polymer conjugates and a STING agonist DMXAA	Subcutaneous	B16-F10 melanoma murine model	- Nanovaccine combined with anti-PD1 treatment led to 50% survival rate on day 38, compared to 20% in mice receiving non-formulated neoantigen peptides.	[102]
exoSTING	Engineered exosome therapeutic that delivers STING agonist to tumor resident APCs	Intratumoral	Checkpoint refractory B16-F10 melanoma murine model	- exoSTING is retained within the injected tumor, and does not induce systemic cytokine production. - exoSTING treatment results in significant induction of PD-L1 expression. In combination with PD1 checkpoint blockade, exoSTING shows enhanced anti-tumor efficacy over high-dose free STING agonist.	[103,104]
STACT-TREX1	Inhibitory microRNA to TREX1, introduced into the STACT strain.	Intravenous	CT26 and MC38 colon carcinoma models, and B16-F10 melanoma model	- Tumor-specific colonization of STACT-TREX1, immune correlates consistent with STING activation and CD8^+^ T-cell-dependent immune response. - Potent tumor growth inhibition and complete tumor regressions with STACT-TREX1 monotherapy. Immunity to tumor re-challenge	[82,83]
STING-NPs	Liposomal nanoparticles (NPs) to deliver the STING agonist, cGAMP	Intravenous	Basal-like triple-negative breast cancer (TNBC) murine model	- cGAMP-NPs accumulate within macrophages at the tumor, induce M2 to M1-like phenotype, MHC and co-stimulatory molecule expression, enhanced CD4^+^ and CD8^+^ T cell infiltration, and tumor apoptosis. - Effective tumor suppression achieved in anti-PD-L1 non-responsive tumors. - Induction of durable anti-tumor T cell responses and prevention of secondary tumor development.	[105]

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
