# Peer review of "STING Agonists as Cancer Therapeutics"

_cancers, 2021, doi:10.3390/cancers13112695_

Round 1

Reviewer 1 Report

In this manuscript, the authors reviewed the anti-tumor mechanisms of cGAS/stimu-lator of interferon genes (STING) pathway, including their effect on immune stimulation, ability for vasculature disruption, and role in the development of tertiary lymphoid structures. Additionally, they addressed the development of STING agonist in pre-clinical and clinical studies for cancer therapy, and their combination with other therapeutic approaches, such as immune-checkpoint blockers. Overall, this manuscript is relatively comprehensive and should be beneficial to a wide range of readers. Few suggestions are listed below.

  1. It would benefit the readers if the authors provide a figure describing the STING pathway, and the targets for their agonists.
  2. Figure 1 is missing in the manuscript.
  3. What is the difference between “Non-CDN agonists” group and “Structure not specified” group listed in Table 2?

Author Response

please see the attachment for response to all reviewers comments

Reviewer 2 Report

The review article by Amouzegar et al presents a summary of the role of STING in triggering immune responses and its use in cancer therapies. As its potency in inducing type I interferons and other proinflammatory cytokines, there is a growing interest in exploiting STING agonists for tumour treatment as well as vaccine adjuvants. In addition to the background knowledge of the STING pathway, the manuscript provides abundant and up-to-date information on the ongoing clinical and preclinical trials of STING agonists. I think this manuscript is suitable for publication in Cancers after a minor revision.

Comments:

  1. Figure 1 is missing.
  2. In Section 1.1, the gain-of-function of STING part seems not truly relevant to the cancer therapy theme of this manuscript. Instead, it would better if the authors could elaborate the role of STING in innate and adaptive immunity, and how the STING activation benefits cancer treatment.
  3. Please check the spelling consistency: e.g. (type-I IFN vs type I IFN vs type-1 IFN), (ADU S-100 vs ADU-S100), (SB 11285 vs SB11285), (TAK676 vs TAK-676)….etc.
  4. Unformatted citation on Page 3: (Chelvanambi et al., 2021; Downey et al., 2014; Huang et al., 2013)
  5. Some of the references seem to come from conference abstracts, such as Ref 77-80. A proper citation style should be used.

Author Response

(The authors gave the same response as above.)

Reviewer 3 Report

Good submission. Fig. 1 and Tables 1 and 2 are very good.

many aspects of STING were discussed

Author Response

(The authors gave the same response as above.)

Reviewer 4 Report

Amouzegar and colleagues provided an extensive review of the current identification and clinical/preclinical study of STING agonists as cancer therapeutics. In general, it can represent an interesting advance in this topic, however I have the following major concerns regarding this work.

Point 1 - There is a huge number of reviews in the literature on the same topic, including the following. In my opinion, the authors should highlight their personal point of view and the originality of their work.

  • Motedayen Aval L, Pease JE, Sharma R, Pinato DJ. Challenges and Opportunities in the Clinical Development of STING Agonists for Cancer Immunotherapy. J Clin Med. 2020 Oct 16;9(10):3323. doi: 10.3390/jcm9103323. PMID: 33081170; PMCID: PMC7602874.
  • Su T, Zhang Y, Valerie K, Wang XY, Lin S, Zhu G. STING activation in cancer immunotherapy. Theranostics. 2019;9(25):7759-7771. Published 2019 Oct 15. doi:10.7150/thno.37574
  • Berger G, Marloye M, Lawler SE. Pharmacological Modulation of the STING Pathway for Cancer Immunotherapy. Trends Mol Med. 2019 May;25(5):412-427. doi: 10.1016/j.molmed.2019.02.007. Epub 2019 Mar 15. PMID: 30885429.
  • Flood BA, Higgs EF, Li S, Luke JJ, Gajewski TF. STING pathway agonism as a cancer therapeutic. Immunol Rev. 2019;290(1):24-38. doi:10.1111/imr.12765

Point 2 - The role of STING agonists in enabling the reinitiation of the immune response in non-immunogenic tumors, i.e. in vaccination approaches, has not been adequately addressed. For example, the STINGVAX vaccine (Fu et al . Science Translational Medicine 2015) and similar can be described.

Minor revisions:

  1. Table 2 should be resized to allow correct indication of column titles
  2. Page 18 – The following sentence should be revised “The presence of a human autoimmune phenotype (SAVI) and an ever-expanding pre-clinical clinical literature supports the potential for numerous STING agonist compounds in boosting anti-tumor immunity and enhancing the effects of existing immunotherapies.”
  3. Pag 3 – In the following sentence, there are three references not correctly reported “Remarkably, tumor vascular normalization also results in “immunologic normalization” in the TME, with suppressor/regulatory or pro-angiogenic immune cell subsets (regulatory T cells [Treg], myeloid-derived suppressor cells, M2-macrophages) being replaced by pro-inflammatory T cells, M1-macrophages and immunostimulatory antigen presenting cells (Chelvanambi et al., 2021; Downey et al., 2014; Huang et al., 2013).”

Author Response

(The authors gave the same response as above.)

Reviewer 5 Report

Comment to the author:

      This review focus on the latest developments in STING targeted therapies; and provides an update on the clinical development and application of STING agonists administered alone, or in combination with immune checkpoint blockade or other approaches. The subject of this article fits the scope of the Cancers. The manuscript may be considered for publication in Cancers after a major revision. Prior publication of this manuscript following points needs to be addressed:

  1. At the beginning of this review, you can introduce STING Agonists first.
  2. There is no graph in “Figure 1. Novel STING Agonist Strategies and Agents in Development.” part in this review.
  3. The content arrangement of the article is disorderly:

(1) The “Novel drug delivery platforms” part can be written as a separate module.

(2) In “STING agonists in pre-clinical evaluations” part, the standards of drug classification are not uniform, it will be better to rearrange the contents of this part.

  1. There are also some typos and grammatical errors that need to be corrected. Please check it carefully:

(1) In “Cyclic Dinucleotides” moiety: Paragraph 4, line 1, the “novel STING agonist” should be “a novel STING agonist”.

(2) In “Systemic Delivery of STING agonists” moiety: Paragraph 1, line 8, the “Compound 3” should be “Compound 3”.

Author Response

(The authors gave the same response as above.)

Round 2

Reviewer 4 Report

The authors have satisfactorily revised the manuscript and all my concerns have been addressed.

Reviewer 5 Report

Comment to the author:

This review focus on the latest developments in STING targeted therapies; and provides an update on the clinical development and application of STING agonists administered alone, or in combination with immune checkpoint blockade or other approaches. The subject of this article fits the scope of the Cancers. I have no other suggestions now,the manuscript may be considered for publication in Cancers in present form.